# Detecting protein and DNA/RNA structures in cryo-EM maps of intermediate resolution using deep learning

Xiao Wang [1], Eman Alnabati[1], Tunde W. Aderinwale[1], Sai Raghavendra Maddhuri Venkata Subramaniya[1], Genki Terashi [2] & Daisuke Kihara [2,1✉]

An increasing number of density maps of macromolecular structures, including proteins and DNA/RNA complexes, have been determined by cryo-electron microscopy (cryo-EM). Although lately maps at a near-atomic resolution are routinely reported, there are still substantial fractions of maps determined at intermediate or low resolutions, where extracting structure information is not trivial. Here, we report a new computational method, Emap2sec+, which identifies DNA or RNA as well as the secondary structures of proteins in cryo-EM maps of 5 to 10 Å resolution. Emap2sec+ employs the deep Residual convolutional neural network. Emap2sec+ assigns structural labels with associated probabilities at each voxel in a cryo-EM map, which will help structure modeling in an EM map. Emap2sec+ showed stable and high assignment accuracy for nucleotides in low resolution maps and improved performance for protein secondary structure assignments than its earlier version when tested on simulated and experimental maps.

[1] Department of Computer Science, Purdue University, West Lafayette, IN, USA. [2] Department of Biological Sciences, Purdue University, West Lafayette, IN, USA. ✉email: dkihara@purdue.edu

Recent years have witnessed rapid advances in the structural determination of biological molecules using cryo-electron microscopy (cryo-EM)[1,2]. The number of determined cryo-EM maps deposited in the public database, EMDB[3], is growing exponentially; and moreover, the fraction of high resolution (e.g., better than 4 Å) maps among them shows a steady increase. Despite the remarkable progress of cryo-EM, there is still a substantial fraction of maps determined at intermediate or low resolutions. Different factors control the achieved resolution of EM maps including conformational or compositional heterogeneity of EM samples, noise level introduced by low electron doses, and inaccurate alignment of the two-dimensional particle images[4].

Depending on the resolution of an EM map, structural information that can be extracted from the map and computational tools appropriate for the task will apparently differ[5]. For maps at a near-atomic resolution (better than 3 Å) or at the subsequent level of the resolution (~4 Å), full atom structure models can be usually built using tools for atomic structure modeling[6–8], de novo main-chain tracing[9,10], structure refinement[11,12] or combinations of them. As the resolution becomes worse, extracting structure information becomes more challenging. In a map of intermediate resolution (~4–10 Å), protein secondary structure elements can be often detected even in cases where tracing a full sequence is difficult. Tools for this task include those which detect typical local densities that correspond to an α-helix and β-sheet[13,14]. Recent methods benefit from the strong image recognition capabilities of deep learning to extract the local and global density features of EM maps[15,16]. Identified structural fragments in a map can be used as clues for tracing a full-length protein chain, to identify known structures from the database that agree with the fragments[17], or to identify structural domains of a complex in the map by their secondary structure content.

Previously, we have developed a method named Emap2sec[16] for detecting protein secondary structure elements in EM maps of intermediate resolutions. Emap2sec showed promising results and provided a novel approach to the structural interpretation of maps at intermediate resolution. However, Emap2sec's detection was limited only to maps of proteins. Here, we have extended the method to detect both protein secondary structure elements and nucleic acids in EM maps of intermediate resolution. The new method, Emap2sec+, uses a more advanced convolutional neural network architecture, Resnet[18], than its predecessor, Emap2sec, and performs a four-class classification, α-helix, β-strand, coil (other structure type), or DNA/RNA, for each voxel in an EM map. Protein nucleic-acid interactions are the core of many essential biological processes including transcription, translation, cell division, and replication. Despite the importance of investigating structures of protein-nucleic acid interactions, there are not many tools available for DNA/RNA structure modeling. Currently, 1340 EM map entries in EMDB have protein-nucleic acid complexes, which is around 13% of the total number of EM maps at EMDB. Recently, a similar tool, Haruspex[19], was developed, which detects nucleotides and protein secondary structure in EM maps. However, Haruspex is designed for high-resolution EM maps of 4 Å or better; thus the aim of the current tool, Emap2sec+ is very different. Emap2sec+ is more suitable for maps of intermediate resolution by design, as we will show later.

We tested Emap2sec+ on two datasets, a dataset of simulated EM maps from 108 complex structures of proteins and nucleic acids at resolution 6 and 10 Å as well as a dataset of 84 experimental EM maps of a resolution between 5 and 10 Å. Emap2sec+ showed high accuracy for nucleotide detection while maintaining comparable, if not achieving better, protein secondary structure detection performance to Emap2sec.

## Results

**The architecture of Emap2sec+.** To use Emap2sec+, the grid size of an input cryo-EM map needs to be adjusted to 1.0 Å. Emap2sec+ takes an input voxel of $11^3$ Å$^3$ extracted from the input EM map and outputs a detected structure at the center of the voxel, which is either DNA/RNA or a protein secondary structure (α-helices, β-sheets, and what we term 'other structures'). Thus, Emap2sec+ classifies a voxel into four different structural classes. The input voxel is slid with a stride of 2 Å to each of the three orthogonal directions in the map and output is computed at each position after a shift. Since the grid space is 1.0 Å, structure assignment is given to every other grid point in the map.

The deep neural network architecture of Emap2sec+ is illustrated in Fig. 1. Emap2sec+ has two phases: In the first phase, Emap2sec+ performs five independent evaluations for an input voxel (Fig. 1b). Among the five evaluations, four evaluations are from binary classifiers, where each of them outputs a probability that the voxel contains a particular structure class (e.g., a residue in an α-helix). The fifth evaluation is by a multi-class classifier, which outputs four probabilities for the four structure classes (Fig. 1b). We show later that combining binary classifiers with the multi-class classifier performed better than using only the multi-class classifier.

Then, the phase 2 network takes the eight probability values from the five classifiers in the phase 1 network. The size of the input voxel of the phase 2 network is $14^3$ Å$^3$, which is shifted with a stride of 2 Å. Since the phase 1 network has assigned probability values at every other grid point, the phase 2 input voxel contains probability values of $7^3$ grid points. The phase 2 network outputs the final probability values of the four structure classes for the center grid point of a query voxel (Fig. 1b). Since the phase 1 and 2 networks use the same stride of 2 Å, phase 2 overwrites results by phase 1. The purpose of this network is to consider detected structures of neighboring voxels in making the decision of the query voxel, which consequently has a smoothing effect on the structural assignment in the EM map.

A detailed architecture of the five classifiers in the phase 1 network is illustrated in Fig. 1c. For an input voxel of size $11^3$ Å$^3$, a convolutional block is applied, which consists of a convolutional layer with 64 filters of size $3^3$ Å$^3$, a 3D batch normalization layer[20] (the batch size is set to 256), and a ReLU activation, which produces 64 voxels of $6^3$ Å$^3$. Then, the voxels are passed to a max-pooling layer with size $2^3$ Å$^3$. After that, each voxel is connected to 6 3D Residual blocks[18,21] (Supplementary Fig. 1) with128, 256, 256, 512, 512, and 1024 filters of size $3^3$ Å$^3$. At the last step, an average pooling layer of size $2^3$ Å$^3$ is applied resulting in a feature vector of 1024 values, which is connected to a fully connected layer to give the final probability values. The phase 2 network incorporates probability values of $7^3$ neighboring voxels. The input is processed with a convolutional block that consists of a 3D convolutional layer with 32 filters of size of 2, a 3D batch normalization layer, and a ReLU activation layer (the batch size is 256), followed by a fully connected network (Fig. 1d). The stride of the convolutional layer was 1. Finally, a softmax operation is applied to the output probability values of the four structure types. The networks were trained on simulated map data for applying to simulated maps while for experimental maps training was performed with experimental maps (Supplementary Table 1). Training details are provided in Methods.

It took about 10–50 min to process one EM map including the time for pre-processing to generate voxel data. The time for structure detection through the phase 1 and 2 networks is proportional to the number of voxels in a map (Supplementary Fig. 2). Users can run Emap2sec+ on a personal computer with

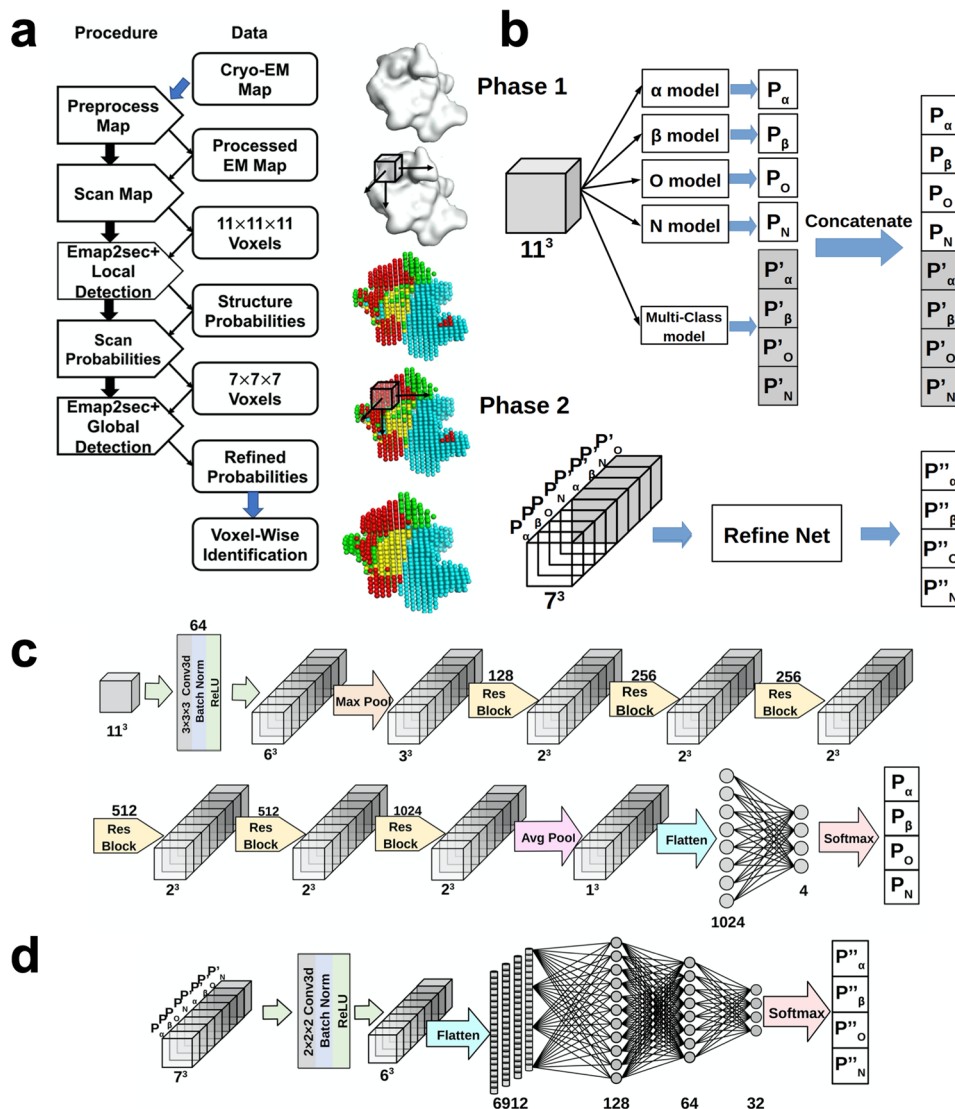

**Fig. 1 The network Architecture of Emap2sec+.** Emap2sec+ scans an EM map with a voxel of 11*11*11 Å$^3$ of size with a stride of 2 and outputs the probabilities that the voxel has α helix, β strand, other structures, or DNA/RNA in the middle of the voxel. It consists of two networks, phase 1 and phase 2, where the phase 2 network refines the initial output by considering assignments given to neighboring 7 × 7 × 7 voxels by the phase 1 network. **a** logical steps of the pipeline. **b** the architecture of phase 1 and phase 2 networks. Phase 1 consists of 4 binary classifiers and one multi (four) -class classifier. The phase 2 network takes outputs from the phase 1 network and outputs refined, final probability values. **c** a detailed network architecture of phase 1. It uses 6 Residual blocks (Supplementary Fig. 1). **d** a detailed architecture of phase 2. The main part is a fully connected network.

GPU. GPU memory size needed for inference is shown in Supplementary Table 2.

Here we formulated the task as a classification problem and used Resnet. However, the problem can be formulated differently such as a segmentation problem with an encoder-decoder framework using U-Net[22] or U-Net+ +[23]. The main reason that we used the classification architecture was because we had the earlier successful experience of Emap2sec[16] with the classification framework. Another reason was because we thought classification would work better than segmentation for intermediate resolution maps.

**Structure detection on simulated maps.** We investigated the performance of Emap2sec+ on a dataset of 108 non-redundant simulated maps each at 6 and 10 Å as well as on a set of 84 experimental maps. The neural networks were trained separately for simulated and experimental maps using independent training

sets. Refer to the Methods section for details of the dataset construction and parameter training of the networks.

The performance of Emap2sec+ was evaluated at three different levels, voxel-based, Q4 (residue/nucleotide-based), and segment-based. F1 score is the harmonic mean of the precision and recall, and a higher value indicates that the performance is good in terms of both precision and recall. Q4 score is defined as the fraction of residues with correctly identified secondary structures of nucleotides the recall for residues in each secondary structure class or nucleotides. In Fig. 2a, voxel-based F1-score and residue-based accuracy are presented in bar graphs for simulated maps at 6 Å and 10 Å. The segment-based accuracy and other result values are provided in Supplementary Table 3. Accuracy values of individual maps of 6 and 10 Å from the phase 1 and 2 networks are made available in Supplementary Data 1.

For the 6 Å maps, Emap2sec+ achieved an overall F1-score of 0.875 and Q4 score of 0.864. Comparing the four structure classes, DNA/RNA had the highest F1-score (0.907) and the

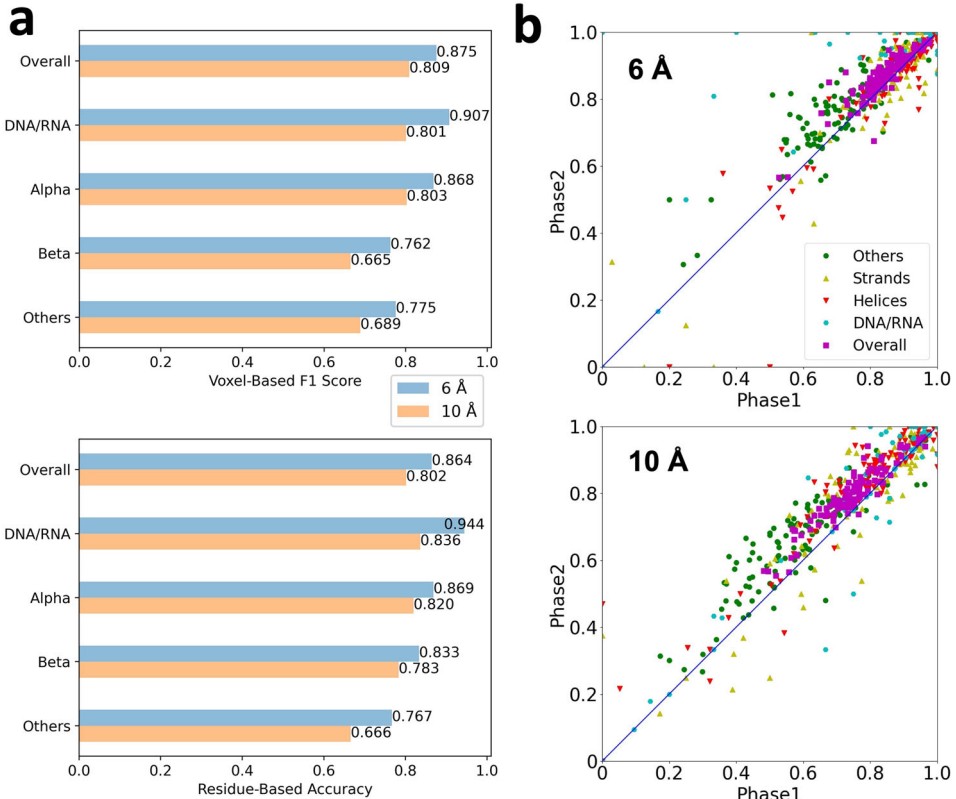

**Fig. 2 The structure detection performance on the simulated map dataset.** The dataset consists of 108 structures computed at two different resolutions, 6 Å and 10 Å. **a** Voxel-based F1 score and Q4 residue/nucleotide-based accuracy for 6 Å (blue) and 10 Å (orange) maps. **b** Comparison of Q4 of phase 1 and phase 2 network outputs for each of 108 test simulated maps computed at 6 Å and 10 Å. Green, other structures; yellow triangles, β strands; red triangles, α helices; cyan, DNA/RNA; magenta, overall Q4.

Q4 score (0.944). The good performance on DNA/RNA detection was not due to a distinct density from proteins. As shown in Supplementary Fig. 3, the density distribution of DNA/RNA has significant overlaps with protein structure classes and is not distinctive particularly in simulated maps. On the other hand, in experimental maps the density of nucleic acids tends to be higher than proteins. Among the three classes in proteins, α helices, β strands, and other structures, α helices were best detected, which is consistent with the previous version of the software, Emap2sec[16]. When only the three protein structure classes are considered, the Q3 score were 0.846 and 0.780 for 6 Å and 10 Å maps, respectively (Supplementary Data 1). These values are comparable with the results shown in Table 1 of the Emap2sec paper[16] (Q3 of 0.831 and 0.798 for 6 Å and 10 Å maps, respectively). Thus, adding the additional class of DNA/RNA in Emap2sec+ did not deteriorate classification for protein secondary structure classes.

Comparing voxel-based accuracy (Supplementary Table 3) and Q4, Q4 was higher in all structural classes. Considering that the residue/nucleotide-based assignments are made by a majority vote from voxels, it indicates that neighboring voxels tend to have consistent assignments, which facilitate users to identify protein and DNA/RNA structures visually in density maps. Also, we noted that the segment-based accuracy of α helices and β strands were very high, 0.950 and 0.940 for 6 Å maps, respectively (Supplementary Table 3). These results strongly indicate that the structure assignments by Emap2sec+ will be able to help main-chain tracing and domain structure assignments in cryo-EM maps.

Results for the 10 Å maps were about 6–13% worse than 6 Å maps (Fig. 2a). Among the four structural classes, α helices have

the smallest difference, 7.5% in terms of the voxel-based F1 score and 5.6% for Q4, between 6 Å and 10 Å maps. Despite the drop, it is remarkable that the overall voxel-based F1 score and Q4 for 10 Å were maintained as high as 0.8. Therefore, we can conclude that there is rich structural information in even 10 Å simulated maps. In Fig. 2b, we compared Q4 from the phase 1 network and the final assignments from the phase 2 network for all 108 maps in the testing set. As shown, overall Q4 values improved for almost all the maps (106 maps, 98.1%) for both 6 and 10 Å resolutions with a margin of 4.6% points and 1.5% points, respectively. There are two maps, 1CA6 and 5GZB, whose Q4 dropped by the phase 2 network. But the margins of the decrease were small: For 1CA6, Q4 dropped by 0.0309 and 0.0273 for the 6 Å map and 10 Å map, respectively. For 5GZB, Q4 decreased by 0.00877 and 0.0247 for 6 Å and 10 Å map, respectively. These drops were caused by incorrect re-assignment of structure class between coil and α helix at the end of an α helix. The change made incorrectly by the phase 2 network for a map was up to two residues.

Among the four structural classes, assignments of other structures improved for the largest fraction of maps (93.5%) for 6 Å map dataset and α helices for 10 Å maps (85.2%). For DNA/RNA class, improvement by phase 2 was observed for 77.7% of maps. As shown in Supplementary Table 3, the phase 2 network made consistent improvement for all the voxel-, residue/nucleotide-, and segment-based metrics for all the structural classes.

In Supplementary Table 3, we also computed the assignment accuracy of another dataset of maps simulated at a randomly selected resolution between 6 and 10 Å. The accuracy of this dataset naturally turned out to be valued between results of 6 Å maps and 10 Å maps.

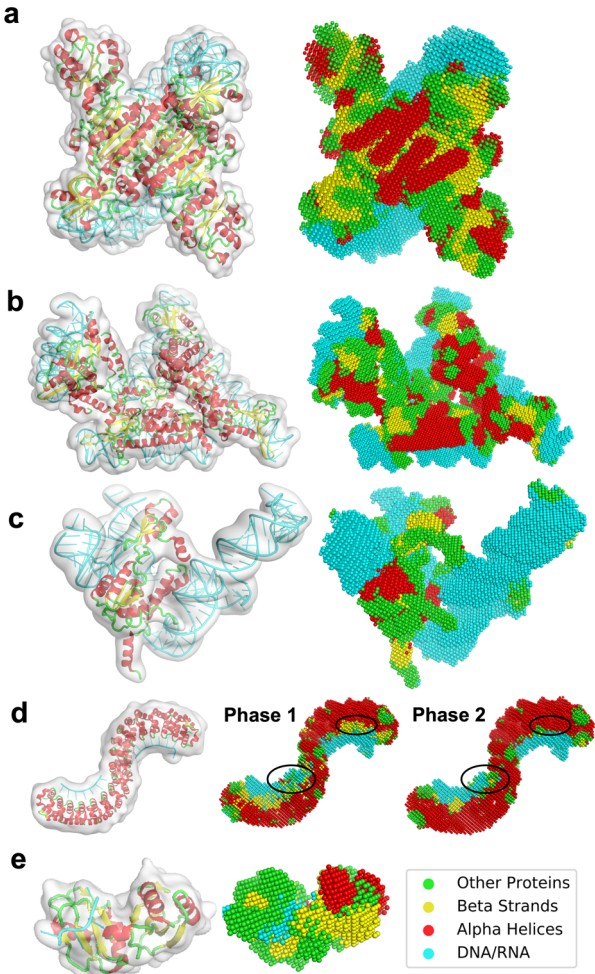

**Fig. 3 Example of the structure detection for simulated maps.** For each panel, the macromolecular structure in the simulated EM map is shown on the left while the structure detection result of the phase 2 network is shown on the right. Colors of spheres in the structure detection panels indicate structure types: red α helices; yellow, β strands; green, other structures (loop); and cyan, RNA/DNA. Detailed evaluation metrics are included in Supplementary Data 1. **a** Aspartyl-tRNA synthase complexed with tRNA (Asp) (PDB ID: 1IL2. Simulated map resolution: 6 Å. The complex contains 1170 amino acids (AA) and 129 nucleotides (nt). Voxel-based F1 score (F1): 0.879; Voxel-based accuracy (Acc): 0.880; Q4: 0.842. **b** Large serine recombinase (LSR) – DNA complex (PDB ID: 4KIS. Simulated at 7.56 Å. 1216 AA and 208 nt. F1: 0.864; Acc: 0.863; Q4: 0.867. **c** Ribosomal protein L30, L37a, S13 complexed with 3 ribosomal RNAs (PDB ID: 1YSH. Simulated Resolution: 10 Å. 261 AA and 163 nt. F1: 0.818; Acc: 0.800; Q4: 0.770. **d** Pumilio homology domain complexed with RNA (PDB ID: 1M8X. Simulated resolution: 10 Å. 682 AA and 15 nt. Phase 1 results: F1: 0.806; Acc: 0.780; Q4: 0.793. Phase 2 results: F1: 0.858; Acc: 0.861; Q4: 0.941. **e** IMP3 RRM12 in complex with RNA (PDB ID: 6GX6. Simulated resolution: 6 Å; 170 AA and 4 nt. F1: 0.712; Acc: 0.715; Q4: 0.753. Accuracies for RNA were: F1(RNA): 0.416; Acc(RNA): 0.270; Q4(RNA): 0.50.

locating in front of this complex were accurately and distinctively detected. The next example, the LSR-DNA complex (Fig. 3b) is abundant in nucleotides and α helical residues. Besides these two abundant structural classes that were detected with a high Q4 of 0.976 and 0.890, respectively. The accuracy of β strands was lower, at Q4 of 0.670, probably mainly because they share only 10.3% of atoms in the complex. In Fig. 3c, a 10 Å map for a ribosomal protein and RNA complex is shown. The large volume RNA structures in this map were well identified at Q4 of 0.877 despite the low resolution. Secondary structures of proteins were detected at a slightly lower residue-based accuracy of 0.703. Figure 3d illustrates changes of structural assignment made by the phase 2 network using a 10 Å map as an example. This protein has a horseshoe fold that has many repeats of α helical structural motifs and contains only very small number of β-strand residues. The phase 1 result of the structure detection contains many small regions with incorrect β-strand detections that are scattered across the map. It is also observed that structure assignments are fragmented as shown in the regions indicated by circles. In contrast, the phase 2 network has smoothened and reduced noise in the structural assignments by considering assignments in neighboring voxels. The phase 2 modification improved the voxel-based F1 score from 0.806 to 0.858 and the Q4 drastically from 0.793 to 0.941. The last panel, Fig. 3e is an example where Emap2sec+ did not perform well. This map, simulated at 6 Å resolution, contains a short RNA of 4 nucleotides. As shown in the right panel, due to the relatively small volume of RNA region, about half of the RNA region was mis-detected as the other structures (green) as indicated by a Q4 of RNA of 0.5, because the RNA fragment is surrounded by loops. In this case, the phase 2 network made the detection of RNA worse by changing the correct assignment of RNA to other structures, resulting in the voxel-based accuracy of nucleic acids reduced from 0.31 to 0.27.

**Structure detection on experimental maps.** Next, we examined the performance of Emap2sec+ on 19 non-redundant experimental maps that are determined at a resolution between 5 and 10 Å. The dataset has six maps between 5 and 6 Å, 7 maps between 6 and 7 Å, 1 map between 7 and 8 Å, 3 maps between 8 and 9 Å, 1 map between 9 and 10 Å, and 1 map at 10 Å. The networks were trained on a dataset of 84 experimental EM maps. The list of EM maps in the dataset is provided in Supplementary Table 1.

Q4 accuracy of individual maps are shown in Fig. 4a relative to their map resolution. The overall Q4 (magenta) exhibits a weak negative correlation to the map resolution. More precisely, when the map resolution is better than 7 Å, more than half of the maps showed relatively high Q4 above 0.64 or higher. For the rest of the maps with a resolution over 7 Å, the average Q4 was 0.464. It is worthwhile to note that Emap2sec+ could detect structures from the map at 10.0 Å (EMD-8131) with Q4 of 0.516, i.e., structures of about half of the residues/nucleotides were correctly identified. Q4 values of the maps are certainly lower than simulated maps that showed over a Q4 of 0.8 on average for even 10 Å maps (Fig. 2), indicating that structure detection is more difficult in experimental maps.

In Fig. 4a, we also showed Q4 value of DNA/RNA (cyan circles). For 14 out of 19 maps, nucleic acids were better detected than the overall average Q4. Interestingly, the detection of nucleic acids did not deteriorate as the resolution was lowered. A high Q4 for nucleic acids of over 0.8 was observed even for maps of over 8 Å resolution.

Turning our attention now to Fig. 4b, as also observed in simulated maps, the phase 2 network improved the accuracy over phase 1 for almost all the maps. Among the four structural classes, α helices and β strands were improved with the largest

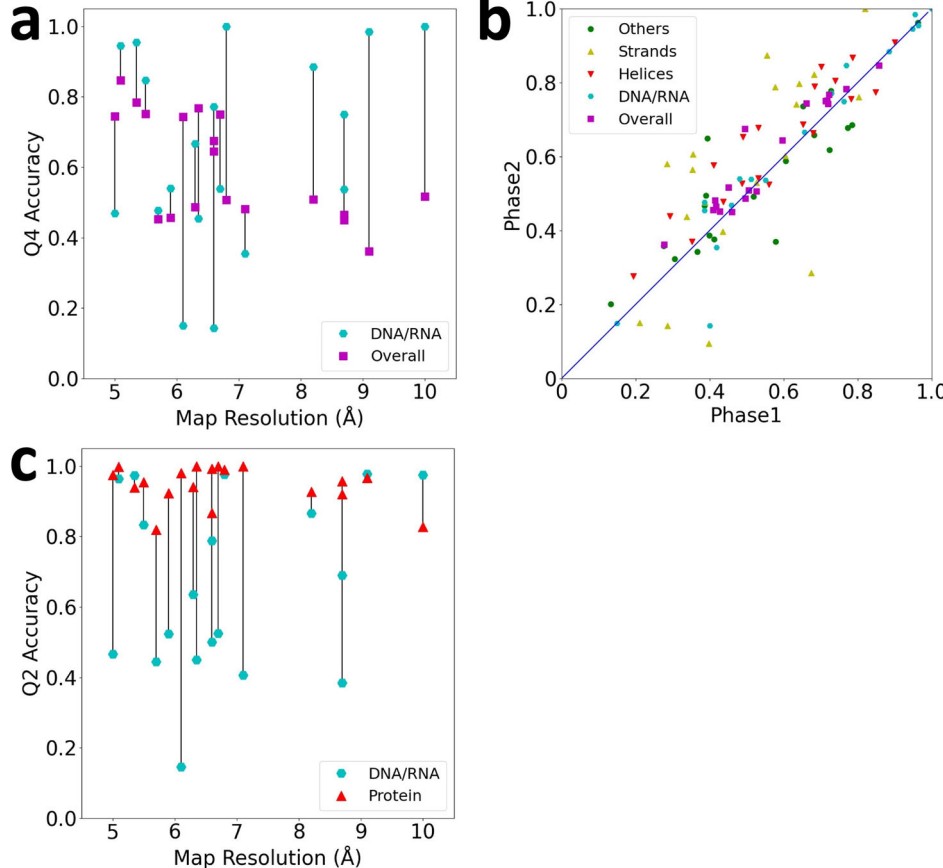

**Fig. 4 Structure class detection on 19 experimental maps.** See Supplementary Data 2 for the phase 1 and phase 2 accuracy of individual maps. **a** Q4 accuracy of experimental maps relative to the map resolution. Overall Q4 is shown in magenta squares and Q4 of DNA/RNA is shown in cyan circles. Lines connect values of the same map. **b** The residue-based accuracy comparison between the Phase 1 and Phase 2 networks. **c** Q2 binary classification accuracy for distinguishing the protein and DNA/RNA classes in experimental maps. Note that values for DNA/RNA can be different from panel a, which reports the results of four-class classification. Since the probability of the protein class was computed as the sum of probabilities of three secondary structure classes, a DNA/RNA assignment in the four-class classification can be changed to protein in the binary classification. Results of individual maps are provided in Supplementary Data 2.

average margin of 0.058 and 0.057, respectively. Though the detection accuracy decreased for other structures for more than half of the maps, overall Q4 improved for all but four maps with an average improvement from 0.56 to 0.60. We also examined the phase 1 accuracy of experimental maps by only using the multi-class classifier (the multi-class model in Fig. 1b). The results are shown in Supplementary Data 3. The results are overall worse than the current combination of the binary and the multi-class classifiers in all the metrics other than the β-sheet voxel-based, residue-based, and segment-based accuracy values.

We further tested the performance of binary classification between the protein class and the DNA/RNA class. The protein class includes all three protein secondary structure classes. As shown in Fig. 4c, all the maps had high residue-based accuracy for the protein class of over 0.8 with an average of 0.946.

We discuss illustrative examples of Emap2sec+'s performance on experimental maps in Fig. 5. The first example (Fig. 5a) is a nucleosome where double-stranded DNA wraps around histones (EMD-3949 PDB ID: 6ESH. The map was determined at 5.1 Å. As shown in the figure, DNA was very clearly identified with a high Q4 of 0.945. α helices and other structures, which dominate histones, were also detected with high accuracy. Q4 of α helices was 0.756, and 0.962 for other structures. As shown in the figure, α helices in histones are distinctively identified to the level that individual helix shapes are visually identified. On the other hand,

β strand residue detection for this map was very low, 0.143. This occurred because the protein chains in this map only contain 34 β strand residues in total, which are isolated into two-residue-long β-strands that do not pair with other β strands to form β sheets. The second map (Fig. 5b) is an example of an RNA-rich structure, the translation pre-initiation complex (EMD-4075, PDB ID: 5LMP. RNA structures are detected with a high Q4 of 0.954. Also, β strands in this map were identified well yielding 0.761 Q4 accuracy. α helices were underpredicted, which resulted in a relatively low Q4 of 0.478. Among 793 α helical residues, 282 (35.6%) were misclassified as other structures, which is the largest protein class in this structure. The next map (Fig. 5c) is opposite to the previous one, whose volume is dominated by proteins with a DNA double-strand embedded in the middle of the structure. It is a 6.35 Å map of the dihedral oligomeric complex with four gyrase A dimers (EMD-9316, PDB ID: 6N1P). The complex holds a double-stranded DNA in the middle. Secondary structures of proteins were identified with high accuracy, particularly for α helices (Q4: 0.843), in this map. Rod shapes of a number of α helices are precisely identified. DNA structure is also detected accurately at the middle of the double strands, but the two ends were mis-recognized as loops and β strands due to lower density values than the middle part. In the last panel (Fig. 5d), we show the structure detection for an 8.7 Å map of TFIID-IIA complex with a promotor DNA. As shown, the DNA is clearly detected,

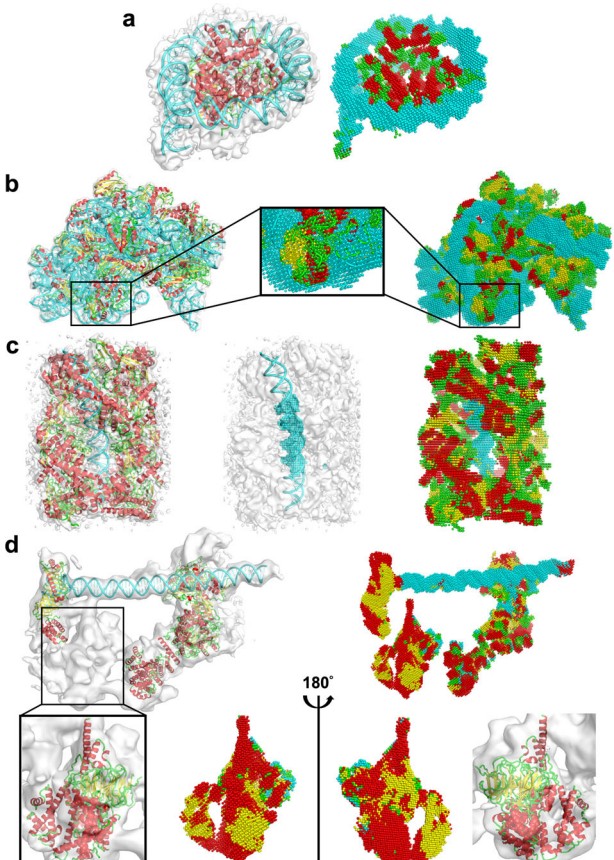

**Fig. 5 Examples of structure detection of experimental maps.** The density maps and associated structures are shown on the left and the detection results of Emap2sec+ are shown on the right. Spheres in red represent detected α helices; yellow, β strands; green, other structures; and cyan, RNA/DNA. Detailed evaluation metrics are shown in Supplementary Data 2. **a** nucleosome breathing Class 3. EMD-3949; 6ESH 10.2210/pdb6ESH/pdb. Resolution: 5.10 Å. 738 amino acids (aa) and 274 nucleotides (nt). Voxel-based F1 score: 0.887; Voxel-based accuracy (Acc): 0.870; Q4: 0.846. **b** bacterial 30S-IF1-IF3-mRNA translation pre-initiation complex. EMD-4075; 5LMP 10.2210/pdb5LMP/pdb. Res.: 5.35 Å. 2622 aa and 1534 nt. F1: 0.855; Acc: 0.846; Q4: 0.784. **c** dihedral oligomeric complex gyrA. EMD-9316; 6N1P 10.2210/pdb6N1P/pdb. Res.: 6.35 Å. 3828 aa and 88 nt. F1: 0.760; Acc: 0.749; Q4: 0.767. In the middle panel, only the DNA is shown with voxels as DNA. **d**. human TFIID-IIA bound to core promoter DNA. EMD-3305; 5FUR 10.2210/pdb5FUR/pdb. Resolution: 8.7 Å; 1857 aa and 39 nt. In a box, another PDB entry, 6MZC 10.2210/pdb6MZC/pdb, is shown, which is for the TFIID BC core and fills the missing structure in lobe B. 6MZC 10.2210/pdb6MZC/pdb was associated with another newer EM map, EMD-9298, determined at a 4.5 Å resolution. F1: 0.487 (0.371); Acc: 0.493 (0.402); Q4: 0.516 (0.438). In the parentheses, values were shown that were computed only for the part of the structure in 6MZC 10.2210/pdb6MZC/pdb that fill the density (953 aa). The structure of lob B and Emap2sec+'s detection is shown from two opposite angles. The detection results using the newer map, EMD-9298, is provided as Supplementary Fig. 4.

which had a Q4 of 0.75 (only the DNA region). The overall Q4 was 0.464, which is about the average for maps with this resolution. The map did not have structure assignment for the structural region named lobe B partly due to the map resolution[24]. But interestingly, the same group later determined the structure of lobe B with a higher resolution, at 4.5 Å[25], which is shown in the box in the second row of the panel. Compared with the newly determined structure, α helices were detected well

with a high Q4 of 0.609 as visualized in the figure. With the 4.5 Å map, Emap2sec+ detected α helices with a slightly better residue-based accuracy of 0.614 and with 18% higher Q4 of 0.517 (0.438 with the 8.7 Å map) (Supplementary Fig. 4).

**Comparison with related works**. Prior to our work, there are very limited methods developed for detecting structures of both DNA/RNA and protein structures. Popular structure modeling tools originally developed for X-ray crystallography, ARP/wARP (v8.0)[8], Phenix[7], and Brickworx[26], only work for maps that include solely proteins or DNA/RNA. The predecessor of this work, Emap2sec, is designed only for protein secondary structure detection. And as mentioned above, Emap2sec+'s performance on protein structure assignments is better than Emap2sec.

A recent work, Haruspex[19], would be closest to Emap2sec+, as it uses a deep neural network to detect both DNA/RNA and protein structures. However, the purpose of their tool and targeted application is quite different; Haruspex is designed to detect structures in higher resolution maps at 4 Å or better to detect potential errors in structure models built from a cryo-EM or to assist modeling process while Emap2sec+ is to provide structural clue for maps at 6–10 Å where structure information is otherwise not easily detectable. In Supplementary Fig. 5 we compared the voxel-based F1 scores of Haruspex and Emap2sec+ on the experimental map dataset. As these two methods have different designs and purposes this comparison is only to illustrate how they differ in nature and to understand the performance of Emap2sec+.

When the average overall F1 score of all the 19 maps was considered, Emap2sec+ had a higher value of 0.620 than Haruspex, which had 0.488. There was one map where Haruspex had a higher overall F1 score. At the individual structure class level, Emap2sec+ had a higher average F1 score by 0.105, 0.066, 0.197, and 0.043 than Haruspex for α helices, β strands, other structures, and DNA/RNA, respectively. The performance of the two methods were closest in the DNA/RNA class, where Haruspex showed a higher F1 score for 8 out of the 19 maps. Emap2sec+ showed a higher F1 score in 13, 14, 17, 11 maps in terms of α helices, β strands, other structures, and DNA/RNA, respectively.

## Discussion

We reported Emap2sec+, a deep learning-based method that can detect structures in EM maps at intermediate resolution (5–10 Å), which substantially upgraded the previous Emap2sec by enabling detection of nucleic acids and improving protein secondary structure detection accuracy. Nucleotides were particularly well detected and the accuracy did not drop much even for maps with lower resolution. This work is the first to explore the structure information for protein-nucleic acid complexes at this difficult resolution range. The same deep learning strategy will be adopted for detecting other molecules or structures, such as amino acid types, in EM maps. Emap2sec+ will aid structural assignments and modeling with fast and accurate predictions and will be a useful and powerful tool in the era of cryo-EM structural biology.

## Methods

**Dataset of simulated and experimental EM maps**. We prepared two types of datasets, simulated and experimental cryo-EM maps. The simulated EM map dataset was computed from 1052 different PDB entries which contain protein and DNA or RNA structures. This dataset is non-redundant in that any protein pairs from two PDB entries have <25% sequence identity between each other. For each PDB entry, we simulated two EM maps at 6 Å and at 10 Å using the pdb2vol program of the SITUS package[27]. The grid size of the computed density maps was set to 1.0 Å.

We collected experimental maps containing protein and DNA/RNA structures from EMDB[3]. We first chose all cryo-EM maps that are determined at a resolution

between 5 and 10 Å and have corresponding PDB entries. Maps were deleted if corresponding PDB entries have an unknown protein sequence with no amino acid type assignment. To ensure that EM maps and associated PDB structures have sufficient overlap and align properly with each other, we examined the cross-correlation of densities between the map and a simulated map from the PDB entry at the map's resolution. Maps were removed if the cross-correlation was <0.65. The alignment between the map and the associated PDB entry was also manually checked. Applying all these steps resulted in a dataset of 84 EM maps.

To handle the redundancy of the maps, maps that have at least one protein sequence pairs with 35% global sequence identity[28] were clustered together using complete-linkage. This clustering resulted in 19 clusters of EM maps. We split these 19 maps into fourfolds as shown in Supplementary Table 1. For training and testing Emap2sec+, we adopted a cross-testing as follows: When each subset was used as the test set, the rest of the three subsets that were used for training were supplemented with cluster members from the 84 EM maps. This operation guarantees that there is no redundancy between the testing and the training sets; and at the same time, the number of training data was enriched with the cluster members.

The grid size of the maps was unified to 1.0 Å by applying trilinear interpolation of the electron density in the maps. Density values of each map were normalized to [0.0, 1.0] with min-max normalization. For experimental maps, negative density values were set to zero before normalization and the density value of the author-recommended contour level was used as the minimum value. For comparison, we also normalized density values of a map using the minimum density value in the entire map and trained the deep neural network and tested. The results were worse on average than the cases where the author-recommended contour level was used for density normalization (Supplementary Data 2 in pages noted as phaseX_nocontour).

The input density data for Emap2sec+ were voxels of a size of $11^3$ Å$^3$ collected from the simulated and experimental EM map dataset by traversing each map along the three axes with a stride of 2 Å. For each voxel, the correct structure(s) was assigned by considering the structures of heavy atoms that were within 3 Å to the center of the voxel. A voxel was considered as outside of the contour and was not taken for input if the voxel center did not have a heavy atom within this range.

For protein heavy atoms, the secondary structure was assigned according to STRIDE[29]. Residues with a label of H, G, or I by STRIDE were labeled as α helix, while β strand was assigned for the labels of B/b or E. Residues with all the other labels were considered as other structures. If a heavy atom belonged to DNA or RNA, the voxel was labeled as DNA/RNA.

**Training the deep neural network of Emap2sec+.** Emap2sec+ was trained separately for the set of simulated maps at 6 Å, the set at 10 Å, and the experimental maps. The networks for simulated and experimental maps were trained with different training sets because the two types of maps have different nature (Supplementary Fig. 3). In our previous work of Emap2sec[16], we tested a network trained on a combined training set with simulated and experimental maps, which did not perform well. For simulated maps, the dataset of 6 Å or 10 Å with 1052 simulated maps was separated into three sets, 844 maps for training and validation for the phase 1 networks, 100 maps for the phase 2 network, and the rest of 108 maps for testing. For the phase 1 networks, 844 maps were split to 80% (675 maps) and 20% (169 maps) for training and validation. The same split of 80 and 20% were applied to the 100 maps for the phase 2 network. The phase 1 networks consist of four binary classification models and a four-class classification model (Fig. 1b). For the binary models, 128 positive and negative voxels each were sampled for a batch (thus the batch size is 256) from the 675 maps, which totals over ten million voxels in an epoch in training. Positive/negative voxels are those which have/do not have the particular structure of the binary classification. For the multi-class model, 64 voxels for each structural class were sampled for a batch (i.e., the batch size is 256 = 64 × 4) from the same 675 maps, which totals over 8.5 million voxels used in an epoch for training. For training the phase 2 network, we used the 100 maps, which do not have overlap with the maps used for phase 1 training and validation. After the phase 1 networks were fully trained, we input the 100 maps and obtained structure assignments to each voxels of the maps. Then, 80 maps (i.e., 80%) were used for training and 20% were used for validation. The number of training voxels for phase 2 was around 1 million. We ran training for 30 epochs.

For training the phase 1 and 2 networks, we tested learning rates of [0.00002, 0.0002, 0.002, 0.02, 0.2] using the Adam optimizer[30] with L2 regularization with values of [1e−6, 1e−5, 1e−4, 0.001, 0.01, 0.1]. Among the combinations tested, the learning rate of 0.002 with L2 regularization parameter of 1e−5 showed the largest voxel-based accuracy on the validation set, although the differences among the combinations were small.

For training the networks for experimental maps, we used the same hyper-parameters as determined for the simulated map datasets. For evaluation 4-fold cross-validation was performed. Similar to the procedure applied for the networks for simulated maps, 80% of the maps in the training set were used for training and validating the phase 1 networks and the remaining 20% of the maps were used for training and validating the phase 2 network. The training for experimental maps took about three days for the phase 1 network using around 3 million voxel input data and about one day for the phase 2 network using about 0.7 million voxel data. We used NVIDIA GTX 2080 GPU for training.

**Evaluation metrics.** We evaluated the performance of Emap2sec+ at three levels: the voxel, the amino acid residue/nucleotide, and the protein secondary structure segment levels. The voxel-level considers if the assigned structure by Emap2sec+ agrees with the structure(s) within a 3 Å radius to the voxel center. We also evaluated with multiple structure labels for a voxel if the structures were observed within the radius. For the voxel-level evaluation, we used the F1-score, which is computed from the precision and the recall of the assignments given to the entire map or to each secondary structure class:

$$F1 - \text{score} = 2 * \frac{\text{precision} \times \text{recall}}{\text{precision} + \text{recall}}, \qquad (1)$$

where precision is the fraction of voxels with correctly assigned structure among all the voxels that were structure-assigned by Emap2sec+ and recall is the fraction of the voxels with correctly assigned structure among all the voxels that belong to the structure class. The overall accuracy (recall) and F1-score of maps were computed as the weighted average of different classes. For the residue-level evaluation, we defined Q4 accuracy. Q4 is the average of the fractions of residues or nucleotides in each structure class whose class label is correctly assigned by Emap2sec+.

For each residue or nucleotide, labels were assigned to voxels within 3.0 Å to any heavy atoms of the residue/nucleotide were considered and the assignment was considered as correct if the majority vote from the voxels agreed with the correct class of the residue/nucleotide. For the protein secondary structures, we further reported the segment-level accuracy. A segment was defined as consecutive amino acids with the same secondary structure type with the minimum length of 6 amino acids for α helix and 3 residues for a β strand. If at least 50% of the residues in a segment were assigned with the correct label, the assignment to the segment was considered correct.

**Reporting summary.** Further information on research design is available in the Nature Research Reporting Summary linked to this article.

## Data availability
The raw data of the structure models built by our method are provided in Supplementary Information, Supp. Table 1 and Data 1. The simulated EM maps can be downloaded from https://doi.org/10.5281/zenodo.4602627. The experimental EM maps can be downloaded from EMDB (https://www.emdataresource.org/). More data that support the findings of this study are available from the corresponding author upon request.

## Code availability
The Emap2sec+ program is freely available for academic use through http://kiharalab.org/emsuites/emap2secplus.php and https://github.com/kiharalab/Emap2secPlus[31].

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

## Acknowledgements

The authors acknowledge Charles Christoffer for his help in finalizing the manuscript. This work was partly supported by the National Institutes of Health (R01GM133840, R01GM123055), the National Science Foundation (DMS1614777, CMMI1825941, MCB1925643, DBI2003635), and the Purdue Institute of Drug Discovery. EA is supported by a fellowship from Umm Al-Qura University, Saudi Arabia.

## Author contributions

D.K. conceived the study. X.W. designed the Emap2sec+ framework with D.K., E.A. and T.W.A. and X.W. implemented it. The datasets were selected and analyzed by E.A. and T.W.A. A script to scan an EM map with a voxel was written by G.T. and a script for visualizing detected structures was implemented by S.R.M.V.S. The experiments were designed by X.W., E.A., T.W.A., and D.K. and were carried out by X.W. X.W. and E.A., S.R.M.V.S., G.T., and D.K. analyzed the results. The manuscript was drafted by X.W. D.K. administrated the project and edited the manuscript.

## Competing interests

The authors declare no competing interests.
