## [Peer Review File · Nature Communications]

REVIEWER COMMENTS

Reviewer #1 (Remarks to the Author):

The manuscript by Wang et al. presents the Emap2sec+ method to assign protein secondary structure types (alpha helix, beta sheet, loop) as well as rna/dna in medium to low-resolution (5-10A) cryoEM density maps using a deep learning approach. Especially identifying DNA/RNA in medium resolution maps is a unique feature.

The work is an advancement of their previously published method Emap2sec (Nat. Methods 16: 911–917(2019)), but is conceptionally very similar, except that the new approach (Emap2sec+) uses a different neural network architecture (Resnet).

The performance of Emap2sec+ is overall only marginally better than Emap2sec for protein secondary structure, however, the main new feature is the capability of Emap2sec+ to detect RNA/DNA.

The method is very useful to interpret medium to low-resolution density maps obtained from cryo-EM in that it annotates of the map as protein secondary structure (alpha/beta) or DNA/RNA. The number of medium to low-resolution cryo-EM maps is increasing quickly, this tool therefore will find an interested audience.

The novelty of this approach is however quite limited.

The work seems technically very well done and overall the method and results are well described.

There are a few issues that should be addressed:

- p7: Figure 2c: it is unclear why a subfigure about experimental maps is shown in a figure that is about simulated maps. These experimental data are not discussed or compared with the results from the simulated maps. So, either omit or discuss differences between simulated and experimental maps at this point.

- p8: It would be helpful if the used scores (F1, Q4) would be explained shortly (not only in the Methods section) before talking about them, at least qualitatively. The definition of Q4 is not clear to me even from the Methods section.

- I assume the analysis of a density map with Emap2sec+ is fast, but a brief discussion on runtime and its dependence on map size would be very interesting.

- Methods: Explain why simulated and experimental maps were trained with different training sets or discuss the influence of the training set in the Discussion.

Reviewer #2 (Remarks to the Author):

KEY RESULTS:

The paper reports a deep learning approach to extracting secondary structure and DNA/RNA information from electron density maps of intermediate resolution (5-10A). The analysis is performed with a set of deep residual CNNs resulting in probability maps for each class. The method was tested on simulated and experimental data. For simulated maps, the method showed good per-nucleotide and per-residue classification accuracy reported at 0.86, of which the latter (per-residue) is comparable with the previous iteration of this work. However, this did not translate to experimental data, where Q3 and Q4 scores were reported at 0.58 and 0.6 respectively, scoring better for individual classes rather than a whole sample, rendering the method less useful for experimental data. Impressively, even with low secondary structure scores in experimental maps, per-nucleotide scores remained very good in low resolution maps, which suggests this method could be perhaps more useful in the context of protein vs. D/RNA

segmentation (a two-class rather than a 4-class problem). Also training multiple models for bins of varying resolutions in experimental data could improve the performance (similarly to two independently trained models in simulated data).

SIGNIFICANCE:

The key value of this method comes from its potential to interpret secondary structure and nucleic-acid elements in electron density maps of intermediate resolution. The ability to perform this analysis on intermediate resolution maps is important to the cryo-EM community, as, despite the so-called 'resolution revolution' in cryo-EM, only ~30% of publicly available EMDB maps have a resolution of 4Å or better. While the currently available automatic model building tools are consistently unable to perform well above 4Å, this method has the potential to aid some of such tools, or give cues to users during model validation.

However, even with promising results in simulated maps, experimental results report the overall average per-class (Q4) accuracy at 0.6, typically scoring much lower for selected classes within sample, which currently renders the method less useful for experimental data. The demonstrated relative high discriminative power of nucleotide vs. protein in low resolution maps is, however, promising. In addition, certain technical ambiguities need to be addressed prior to publication (see more comments in later sections of this review). Furthermore, the presentation and language of the paper require improvement. The novelty of this work lies in its application to difficult low resolution data, which is notoriously challenging for an automated analysis. It also presents a novel approach to analysing secondary structure in electron density maps by utilising ResNet architecture, which to my knowledge has not been attempted before on this type of data.

DATA / METHODOLOGY / IMPROVEMENTS

Overall, the authors seem to approach the training as a classification problem, with map subregions as the network input and a single value output (for the central voxel of the subregion). This approach seems inefficient, as the evaluation of large number subregions is required (even with a stride of 2). A much more light-weight approach could be to treat this as a segmentation problem with encoder-decoder architecture, where the output of the network is a voxel-wise classification for each voxel in the input region (which is a more standard way also adopted by other papers in the field e.g. Haruspex). This would not only allow the authors to perform sampling of the map with a much larger stride, but also as a consequence, reduce the computational and time complexity. Additionally, these networks tend to produce much smoother segmentations, as the classifications already incorporate the neighboring information (which could remove the need for the phase 2 network). The authors do not provide any commentary on whether they considered encoder-decoder architectures for this problem. At the very least, the authors should mention this alternative, or explain why it was unsuitable to the problem when comparing to related work. The previous version of the Emap2sec paper also doesn't seem to mention whether such alternative architectures were considered.

Additionally, aside from training, I am also concerned that this approach would produce very long evaluation times, which, from the user perspective, could render this method less useful. The authors do not appear to comment on training or evaluation times. It would also be useful to read more details of the training setup (e.g. what hardware was it trained on, requirements for user inference).

The results for synthetic structures are very good (overall per-voxel F1 0.87 and Q4 accuracy 0.86 for 6A). However, the reported F1 score, accuracy, Q3 and Q4 in experimental maps were .61, .60, .58 and .59 respectively (Supplementary Table 4). In particular, the reported per-class (Q4) accuracy for coil, beta, alpha and DRNA classes are .5, .50, .55 and .65 respectively, suggesting the method is less useful for protein (as opposed to nucleotide) data. Half of the experimental data (9 out of 19 experimental maps) is reported as under 50% Q3 accuracy and just under half (7 out of 19) is reported as under 50% on Q4 accuracy. Isolated DRNA predictions were much better, notably 5 samples (7290, 4671, 3949, 4075 and 8131) reported both, per-nucleotide recall and per-voxel accuracy, of over .9, impressively, even in the maps with resolution as low as 10Å. This suggests that the method could be more useful for a two-class segmentation problem of nucleotide

vs. protein segmentation (i.e. without the secondary structure information).

In cryo-EM maps, the distribution of background is usually expected to be much higher than the other classes. The authors do not report the influence of background nor how they handled such class imbalance during training (e.g. specialised loss functions, masking etc.).

Grammatical, stylistic, language mistakes and typos are present throughout the paper, e.g. "(...) including proteins and protein and DNA/RNA complexes (...)", or "with a voxels" rather than "with voxels", or " binary classifies" rather than "classifiers", or "(...) beta-strands in this map was identified well (...)" rather than "were" etc. Occasionally there are missing spaces and punctuation. The presentation and language of the paper needs improvement.

ADDITIONAL DETAILED COMMENTS:

Introduction/Methods and Data

It is difficult to understand exactly what the network inputs were. The authors refer to a "cryo-EM map with a voxel of 11^3 \AA^3 with a stride of 2 \AA ", which does not make it clear whether that is the voxel size of the input map, or the input size to the network ($11 \times 11 \times 11$ voxels) as a subsection of the map. This ambiguity is consistent throughout the paper. Later the authors state " 7^3 neighboring voxels", but it is unclear whether this refers to analysing a subregion of the map of $7 \times 7 \times 7$ voxels as the network input, or a map with voxels of size 7^3 \AA . It is possible there is a vocabulary confusion between a 'voxel' and 'region'/'input size'/'kernel receptive field' and it needs clarification. It is also unclear why the filter sizes are given in Angstroms. My understanding is that subregions of the map ($11 \times 11 \times 11$ voxels phase 1 and $7 \times 7 \times 7$ voxels phase 2) were used as the network input, but if that is the case, it was not immediately clear and should be clarified.

It is also difficult to understand exactly the experimental setup of the network/(s). The authors state that during phase 1 the network performs "five independent evaluations", which is unclear as to whether this indicates a single feed-forward pass through a model with a five channel output, five independent feed-forward passes through the same model with different labels, or a single feed-forward pass through five different models. Later, the description of Figure 1 clarifies that five different models (four binary and one multi-class) were used, where the last model output contains multi-channel class probabilities, but it should be made clear from the text. The authors further indicate a five channel output, where the last channel contains four-channel probability values, but it is misleading to refer to independent network outputs as "channels" and this should be clarified in the text. Authors further state that the eight resulting probability values are combined as an input to phase 2, whereas they previously stated that four out of the eight contain binary values, not probabilities. Perhaps it would be better to state that outputs from five independent classifiers, including four binary single-channel outputs and one probability four-channel output, were combined as an input to phase 2.

I'm also not convinced if there are any advantages arising from training four binary classifiers in addition to a single multi-class classifier, as opposed to obtaining four separate binary maps as thresholded probability values from a multi-class classifier's softmax output. The authors seem to split the training data between the five different classifiers in phase 1 (as opposed to training the five classifiers on the same data) which reduces the amount of data each classifier learns from, but the authors provide no justification for such a split. This could be to prevent overfitting, but the authors should elaborate.

In experimental maps, the authors normalize the maps with the minimum value set to the author-recommended contour level, effectively truncating potentially valuable signal. While recommended contour levels are useful for display, such truncated signal, while difficult to interpret by a human, could contain enough discriminative information to drive a machine learning algorithm (which could also explain considerably lower results in experimental maps). It would be useful to comment on this in the paper. I am also concerned that this could be ignoring some of the more noisy, blurry or flexible components of the maps.

Results:

The authors claim that, "When only the three protein structure classes are considered, the Q3 score were 0.846 and 0.780 for 6 Å and 10 Å maps, respectively (Supplementary Table S3), which are better than results shown in Table 1 of the Emap2sec paper." However, the authors do not report the actual values from the previous Emap2sec paper. Referring to Table 1 and Supplementary Figure 3 of their original paper reveals that the overall Emap2sec Q3 scores were actually comparable to Emap2sec+ (+/- 0.02) and in fact slightly lower in Emap2sec+ for 10A (0.780 for Emap2sec+ and 0.798 for Emap2sec).

The authors report 106 out of 108 samples improved following phase 2 training. It would be useful to elaborate which two samples did not improve, and whether there was anything particular to set them apart.

The authors do not mention the distribution of nominal resolution in experimental maps. Investigation of Supplementary Table 4 and Figure 4a reveals that low resolution maps (7-10A) were less frequent. This could perhaps be reported in the text.

The authors state that "It is worthwhile to note that Emap2sec+ could detect structures from the map at 10.0 Å (EMD-8131) with Q4 of 0.575". However, consulting the Supplementary Table 4, EMD-8131 Q4 is reported as 0.516. The authors also claim that "For the rest of the 10 maps [with resolution over 7Å], Q4 was stable around 0.45 to 0.5 regardless of the map resolution". However, it is clear from Figure 4a and Supplementary Table 4 that a single sample was lower than that (EMDB-4671 Q4 at 0.3617). This is understandable given that the map was at 9.1Å resolution, nevertheless the results should be reported correctly in the text.

Text discussion of Figure 5 frequently reports good Q4 on presented experimental structures, however consistently omits the bad predictions (e.g. EMDB-4075 0.47 Alpha recall, EMDB-3949 0.14 Beta recall, as reported by Supplementary Table 4). This is not obvious without consulting the Supplementary Table 4 and could thus bias the reader. In some cases, these lower scores are perhaps understandable given the small proportion of the particular class represented in the structure (e.g. beta-strands in EMDB-3949), nevertheless the negative results should still be reported in text while discussing the particular structure, as this largely determines how useful the method is for experimental data.

Comparison with related works:

Stating that Haruspex did not perform well on simulated maps is inadequate and such comparison should not be made. The fact that it did not perform well for simulated data is obvious by design, as Haruspex was trained for experimental maps. The Supplementary Figure 3 shows the comparison between Haruspex on both 6 and 10Å simulated maps against Emap2sec+, which, as earlier stated, was trained separately for simulated and experimental data (and even made the distinction between 6 and 10Å maps in simulated data). These comparisons should be excluded from the paper.

The comparison made between Haruspex and Emap2sec+ on experimental data is far more sensible. However, given the fact that Haruspex was trained for high (sub 4Å) resolution data, it is interesting to see that many Haruspex predictions were only slightly improved on by Emap2sec+, and in some cases Haruspex was actually better. While the large majority of predictions by Emap2sec+ were indeed better than Haruspex, some of these exceptions could be discussed in text. Also, it would be interesting to read a more comprehensive comparison between the two methods, e.g. average difference in accuracy or average per-class difference in accuracy. For example, from the figure it appears that loops ('others' category) were on average classified much better by Emap2sec+, but for DRNA only 10 out of 19 structures (about half) were better with Emap2sec+ (although I can only count 18 DNA/RNA data points in that figure rather than 19). The text only states "(...) Haruspex did not work well on our low resolution datasets of simulated maps and experimental maps (...)", which is overly dismissive and performs an inadequate comparison.

Responses to comments by Reviewer 1:

The manuscript by Wang et al. presents the Emap2sec+ method to assign protein secondary structure types (alpha helix, beta sheet, loop) as well as rna/dna in medium to low-resolution (5-10A) cryoEM density maps using a deep learning approach. Especially identifying DNA/RNA in medium resolution maps is a unique feature.

The work is an advancement of their previously published method Emap2sec (Nat. Methods 16:911–917(2019)), but is conceptionally very similar, except that the new approach (Emap2sec+) uses a different neural network architecture (Resnet).

The performance of Emap2sec+ is overall only marginally better than Emap2sec for protein secondary structure, however, the main new feature is the capability of Emap2sec+ to detect RNA/DNA.

The method is very useful to interpret medium to low-resolution density maps obtained from cryo-EM in that it annotates of the map as protein secondary structure (alpha/beta) or DNA/RNA.

The number of medium to low-resolution cryo-EM maps is increasing quickly, this tool therefore will find an interested audience.

The novelty of this approach is however quite limited.

The work seems technically very well done and overall the method and results are well described.

Thank you very much for taking the time to review the paper and your overall positive comments. As you mentioned, compared to the previous work, the new method, Emap2sec+ can detect DNA/RNA in EM maps. Technically, in this work we used Resnet, rather than a classical basic convolutional neural network that was used in Emap2sec. Also, we proposed a new 2-phase network architecture, where the phase 1 network combines independent binary and multi-class detection from the first phase, which would be new in 3D structure detection.

There are a few issues that should be addressed:

- p7: Figure 2c: it is unclear why a subfigure about experimental maps is shown in a figure that is about simulated maps. These experimental data are not discussed or compared with the results from the simulated maps. So, either omit or discuss differences between simulated and experimental maps at this point.

We removed Figure 2c from Figure 2 and moved it to the Supplementary Figure S2. By referring to the Suppl. Figure S2, we just mentioned in the text that the differences between simulated and experimental maps as pointed out.

- p8: It would be helpful if the used scores (F1, Q4) would be explained shortly (not only in the Methods section) before talking about them, at least qualitatively. The definition of Q4 is not clear to me even from the Methods section.

We have added the description of F1 and Q4 in page 8. Also, we revised the explanation of Q4 in page 20.

- I assume the analysis of a density map with Emap2sec+ is fast, but a brief discussion on runtime and its dependence on map size would be very interesting.

We added a new Supplementary Figure S2 and discussed it in page 5. Maps can be processed overall within 10-50 minutes. Once the preprocessing of maps is done, the time for structure detection in a map simply correlates with the number of voxels in the map (Supplementary Figure S2, panel b).

- Methods: Explain why simulated and experimental maps were trained with different training sets or discuss the influence of the training set in the Discussion.

There are two reasons. First, as shown in Supplementary Figure S2, the density value distributions of simulated maps and experimental maps have noticeable difference. Particularly, density levels of DNA/RNA are similar with those of proteins in simulated maps while DNA/RNAs are different in experimental maps.

Second, in the previous work of Emap2sec (Nature Methods 2019), we did train the deep neural network with a combined training dataset with experimental maps and simulated maps and tested it on the experimental maps. This turned out that the detection accuracy dropped for more than half of the cases. This is mentioned in the main text in the Emap2sec paper (<https://www.nature.com/articles/s41592-019-0500-1>):

The results shown for the experimental maps (Figs. 4 and 5, and Supplementary Table 3) were obtained by using Emap2sec trained on experimental maps. Adding simulated maps in the training set to increase the amount of training data did not yield consistent improvement of detection results. The voxel-based F_1 score improved for 14 maps (32.6%) but deteriorated for 22 maps (51.2%) (with no change for 7 maps). As a consequence, the overall average voxel-based F_1 score slightly deteriorated (Supplementary Fig. 2). Thus, probably because of the different nature of the two types of maps, adding simulated maps in training did not substantially improve structure detection in experimental maps.

As mentioned in the end of the paragraph, this is probably because simulated maps and experimental maps are largely different as the simulated maps are noise-free and uniform while experimental maps have noise and often uneven resolution in local regions in maps.

We added the following sentence in page 19:

“The networks for simulated and experimental maps were trained with different training sets because the two types of maps have different nature (**Supplementary Figure S3**). In our previous work of Emap2sec¹⁶, we tested a network trained on a combined training set with simulated and experimental maps, which did not perform well.”

Responses to comments by Reviewer 2:

KEY RESULTS:

The paper reports a deep learning approach to extracting secondary structure and DNA/RNA information from electron density maps of intermediate resolution (5-10Å). The analysis is performed with a set of deep residual CNNs resulting in probability maps for each class. The method was tested on simulated and experimental data. For simulated maps, the method showed good per-nucleotide and per-residue classification accuracy reported at 0.86, of which the latter (per-residue) is comparable with the previous iteration of this work. However, this did not translate to experimental data, where Q3 and Q4 scores were reported at 0.58 and 0.6 respectively, scoring better for individual classes rather than a whole sample, rendering the method less useful for experimental data. Impressively, even with low secondary structure scores in experimental maps, per-nucleotide scores remained very good in low resolution maps, which suggests this method could be perhaps more useful in the context of protein vs. D/RNA segmentation (a two-class rather than a 4-class problem). Also training multiple models for bins of varying resolutions in experimental data could improve the performance (similarly to two independently trained models in simulated data).

SIGNIFICANCE:

The key value of this method comes from its potential to interpret secondary structure and nucleic-acid elements in electron density maps of intermediate resolution. The ability to perform this analysis on intermediate resolution maps is important to the cryo-EM community, as, despite the so-called 'resolution revolution' in cryo-EM, only ~30% of publicly available EMDB maps have a resolution of 4Å or better. While the currently available automatic model building tools are consistently unable to perform well above 4Å, this method has the potential to aid some of such tools, or give cues to users during model validation.

However, even with promising results in simulated maps, experimental results report the overall average per-class (Q4) accuracy at 0.6, typically scoring much lower for selected classes within sample, which currently renders the method less useful for experimental data. The demonstrated relative high discriminative power of nucleotide vs. protein in low resolution maps is, however, promising. In addition, certain technical ambiguities need to be addressed prior to publication (see more comments in later sections of this review). Furthermore, the presentation and language of the paper require improvement. The novelty of this work lies in its application to difficult low resolution data, which is notoriously challenging for an automated analysis. It also presents a novel approach to analyzing secondary structure in electron density maps by utilizing ResNet architecture, which to my knowledge has not been attempted before on this type of data.

Thank you for the all the constructive comments. Since each point is mentioned separately below, we respond to each of them there.

We worked with a native English speaker, who is mentioned in Acknowledgement, to improve language of the manuscript.

DATA / METHODOLOGY / IMPROVEMENTS

Overall, the authors seem to approach the training as a classification problem, with map subregions as the network input and a single value output (for the central voxel of the subregion). This approach seems inefficient, as the evaluation of large number subregions is required (even with a stride of 2). A much more light-weight approach could be to treat this as a segmentation problem with encoder-decoder architecture, where the output of the network is a voxel-wise classification for each voxel in the input region (which is a more standard way also adopted by other papers in the field e.g. Haruspex). This would not only allow the authors to perform sampling of the map with a much larger stride, but also as a consequence, reduce the computational and time complexity. Additionally, these networks tend to produce much smoother segmentations, as the classifications already incorporate the neighboring information (which could remove the need for the phase 2 network). The authors do not provide any commentary on whether they considered encoder-decoder architectures for this problem. At the very least, the authors should mention this alternative, or explain why it was unsuitable to the problem when comparing to related work. The previous version of the Emap2sec paper also doesn't seem to mention whether such alternative architectures were considered.

Thank you, it is a valid point. In this work we used ResNet to consider this as classification problem as you pointed out. But the problem can be formulated in other ways too. I also agree that another potential approach is to consider it as a segmentation problem as Haruspex did using U-Net.

The reason that we did not choose segmentation but classification is mainly from two considerations:

First, in segmentation, detecting foreground and background is a main focus. Therefore, usually applying a segmentation method is very suitable when the resolution of images are high, where distinction between background and foreground is not very difficult. However, since here we handle low resolution maps of up to 10 A, we thought this is not a typical case for applying segmentation. Thus, we thought formulating the problem as classification would be more suitable, although we have not actually developed a segmentation program and compared it with the current Emap2sec+.

The second reason was because we formulated the task as classification in our previous work of Emap2sec, which yielded reasonable accuracy.

Moreover, instead of U-Net architecture, there are also other choices such as VAE, U-Net++, U-Net3Plus for encoder-decoder architecture.

We mentioned this explanation in page 5.

Additionally, aside from training, I am also concerned that this approach would produce very long evaluation times, which, from the user perspective, could render this method less useful. The authors do not appear to comment on training or evaluation times. It would also be useful to read more details of the training setup (e.g. what hardware was it trained on, requirements for user inference).

We added new data of computational time for inference (evaluation) as Supplementary Figure S3 and mentioned it in page 5. As shown, the computational time including pre-processing steps was 10-50 minutes for an experimental map. If we only consider the computational time of the deep learning part (panel B), it was clearly proportional to the number of voxels.

We mentioned the time for training deep learning networks in page 19. It took about 3 days and 1 day for the phase 1 and phase 2 networks on the experimental map data, respectively.

Both training and evaluation was done using one GPU node of NVIDIA GTX 2080. We only had this GPU available in our group but the computational times would be shorter if better computational resource is available.

The results for synthetic structures are very good (overall per-voxel F1 0.87 and Q4 accuracy 0.86 for 6A). However, the reported F1 score, accuracy, Q3 and Q4 in experimental maps were .61, .60, .58 and .59 respectively (Supplementary Table 4). In particular, the reported per-class (Q4) accuracy for coil, beta, alpha and DRNA classes are .5, .50, .55 and .65 respectively, suggesting the method is less useful for protein (as opposed to nucleotide) data. Half of the experimental data (9 out of 19 experimental maps) is reported as under 50% Q3 accuracy and just under half (7 out of 19) is reported as under 50% on Q4 accuracy. Isolated DRNA predictions were much better, notably 5 samples (7290, 4671, 3949, 4075 and 8131) reported both, per-nucleotide recall and per-voxel accuracy, of over .9, impressively, even in the maps with resolution as low as 10Å. This suggests that the method could be more useful for a two-class segmentation problem of nucleotide vs. protein segmentation (i.e. without the secondary structure information).

Checking binary segmentation between protein and DNA/RNA is great idea. We show the results in the new Figure 4C and a new page in Supplementary Table 4 (Excel file), the third datasheet named “Binary Eval(uation)”. Remarkably, the binary detection showed very good results. The average per-class (Q2) accuracy of protein is now higher than DNA/RNA for many maps, at a very high value of 0.946. This high value is well maintained for maps at 7 Å or worse. We also provided the codes for the binary classification in Github. We are really thankful to the reviewer for the great insight.

In cryo-EM maps, the distribution of background is usually expected to be much higher than the other classes. The authors do not report the influence of background nor how they handled such class imbalance during training (e.g. specialized loss functions, masking etc.).

We added more explanation about the way we took voxels from EM maps in page 19.

Voxels were not taken from background where the center of the voxels does not correspond to proteins or DNA/RNA. (thus, the Emap2sec+ network output probability values of 4 classes, i.e. 3 protein secondary structures and DNA/RNA, but does not output a probability of

background). This is because we assume that users apply Emapsec+ to density regions where they believe a protein or DNA/RNA exists.

Among the four structure classes, to balance the training data, the same number of voxels were collected for training. This design showed better and stable performance compared to using weighted cross entropy loss. We added one sentence in page 19 to clarify this.

Grammatical, stylistic, language mistakes and typos are present throughout the paper, e.g. “(...) including proteins and protein and DNA/RNA complexes (...)”, or “with a voxels” rather than “with voxels”, or “ binary classifies” rather than “classifiers”, or “(...) beta-strands in this map was identified well (...)” rather than “were” etc. Occasionally there are missing spaces and punctuation. The presentation and language of the paper needs improvement.

With a native English speaker’s help, we carefully went over the manuscript to fix typos and grammatical errors.

ADDITIONAL DETAILED COMMENTS:

Introduction/Methods and Data

It is difficult to understand exactly what the network inputs were. The authors refer to a “cryo-EM map with a voxel of 11^3 \AA^3 with a stride of 2 \AA ”, which does not make it clear whether that is the voxel size of the input map, or the input size to the network (11x11x11 voxels) as a subsection of the map. This ambiguity is consistent throughout the paper. Later the authors state “ 7^3 neighboring voxels”, but it is unclear whether this refers to analyzing a subregion of the map of $7 \times 7 \times 7$ voxels as the network input, or a map with voxels of size 7^3 \AA . It is possible there is a vocabulary confusion between a ‘voxel’ and ‘region’/’input size’/’kernel receptive field’ and it needs clarification. It is also unclear why the filter sizes are given in Angstroms. My understanding is that subregions of the map (11x11x11 voxels phase 1 and $7 \times 7 \times 7$ voxels phase 2) were used as the network input, but if that is the case, it was not immediately clear and should be clarified.

We agree that it is confusing. Thank you for pointing out, we should have described it more carefully:

First, the grid size of experimental maps is unified to 1.0 \AA by applying trilinear interpolation of the electron density in the maps. For simulated maps, grid size was set to 1.0 \AA . (This was not mentioned in this manuscript. We added it and in page 4 and in the Method section).

The input data of Emap2sec+ phase 1 network is a voxel with a size of $11 \text{ \AA} \times 11 \text{ \AA} \times 11 \text{ \AA}$, which is a subsection of an EM map. Since now the grid size is unified to 1.0 \AA , this voxel has $11 \times 11 \times 11$ grid points, where density values are assigned. In the phase 1 network, probability values of 4 structural classes from the four binary classifications and one multi-class classification (thus, in total, there are 8 probability values) are computed for the center grid point of an input voxel.

Then, the input voxel is slid by an interval of 2 \AA to each of x, y, z directions, and outputs 8 probability values at each place the voxel is placed. Since the grid size is 1.0 \AA , by sliding the

input voxel, the probability values are assigned to every other grid points (skipping one grid point).

Next, these structure assignments by the phase 1 network are subject to the phase 2 network. The size of the phase 2 input voxel is actually $14 \text{ \AA} \times 14 \text{ \AA} \times 14 \text{ \AA}$, but since the structure assignment in the phase 1 is made every other grid points, the number of grid points that have assigned structure probabilities is $7 \times 7 \times 7$ (This was inaccurate in the original manuscript). So the phase 2 network considers $7 \times 7 \times 7$ grid points with probability assignments spread in the space of $14 \times 14 \times 14 \text{ \AA}^3$, and make the final structure probability computation for the center grid point. Then, the input voxel is shifted by 2 \AA (i.e. by 2 grid points); thus, phase 2 network basically computes probability values of 4 structure classes at the same grid points as the phase 1 network did, overwriting phase 1 structure probabilities.

We now use “voxel” to indicate the input 3D data. And we used “grid point” to indicate the positions in EM maps that have density values assigned. We have made significant changes in page 4 and 5. We also clarified that the grid size was adjusted to 1.0 \AA in the Method section.

It is also difficult to understand exactly the experimental setup of the network/(s). The authors state that during phase 1 the network performs “five independent evaluations”, which is unclear as to whether this indicates a single feed-forward pass through a model with a five channel output, five independent feed-forward passes through the same model with different labels, or a single feed-forward pass through five different models.

There are 5 channels: 4 channels are binary classifications, each of which judges whether an input voxel contains alpha helix or not; beta strand or not, coil (other structure) or not, DNA/RNA or not. Each of which outputs a probability value of having the structure class. The last one is a 4-class classification, which produces a probability value for each of the 4 structural classes. So an input voxel is passed through these 5 channels independently and in parallel.

Later, the description of Figure 1 clarifies that five different models (four binary and one multi-class) were used, where the last model output contains multi-channel class probabilities, but it should be made clear from the text. The authors further indicate a five channel output, where the last channel contains four-channel probability values, but it is misleading to refer to independent network outputs as “channels” and this should be clarified in the text. Authors further state that the eight resulting probability values are combined as an input to phase 2, whereas they previously stated that four out of the eight contain binary values, not probabilities. Perhaps it would be better to state that outputs from five independent classifiers, including four binary single-channel outputs and one probability four-channel output, were combined as an input to phase 2.

As I answered in the previous question, a binary model outputs a probability value that an input voxel contains a structure class (e.g. alpha helix), by performing a binary classification between the structure class (the voxel contains a residue of the structure class, e.g. alpha helix) vs. negative (the voxel does not contains a residue of the structure class). Thus there are 4 probability values from the 4 binary classifications and additional 4 probability values from the 4-class multi-class classification model. We rephrased channel to classifier in the text.

Thank you again for all these comments and questions. We have rewritten the entire section for clarification.

I'm also not convinced if there are any advantages arising from training four binary classifiers in addition to a single multi-class classifier, as opposed to obtaining four separate binary maps as thresholded probability values from a multi-class classifier's softmax output. The authors seem to split the training data between the five different classifiers in phase 1 (as opposed to training the five classifiers on the same data) which reduces the amount of data each classifier learns from, but the authors provide no justification for such a split. This could be to prevent overfitting, but the authors should elaborate.

During the development, we did compare the phase 1 accuracy of only using the multi-class classifier and the current combination of binary & multi-class classifiers. We now added a new table, Supplementary Table S5. The results show that the current setting performs better in all the metrics (columns in the table) except for beta-sheet voxel-based accuracy, residue-based accuracy, and segment-based accuracy. This is mentioned in page 13. We have chosen the current setting because the overall accuracy values, i.e. voxel-based F1, voxel-based accuracy, overall Q3 and Q4 were better by the current setting.

Regarding the training dataset, for both binary classifiers and multi-class classifiers, voxels were taken from the same set of maps. But the entire data used for training for binary and multi-class are not 100% identical, because training voxels are sampled for each batch on-the-fly (batch size = 256) from the same set of maps, and voxels from a smaller class were up-sampled, i.e. some voxels were selected multiple times in one training epoch. We run training for 30 epochs.

We expanded the explanation in page 19.

Also, as you mentioned, combining binary and multi-class classifiers probably help avoiding overfitting because of the optimizing functions are different for binary and multi classifiers. This strategy is also frequently adopted in ensemble methods to boost their performances in competitions, like Kaggle (<https://www.kaggle.com/>).

In experimental maps, the authors normalize the maps with the minimum value set to the author-recommended contour level, effectively truncating potentially valuable signal. While recommended contour levels are useful for display, such truncated signal, while difficult to interpret by a human, could contain enough discriminative information to drive a machine learning algorithm (which could also explain considerably lower results in experimental maps). It would be useful to comment on this in the paper. I am also concerned that this could be ignoring some of the more noisy, blurry or flexible components of the maps.

It is an interesting and valid question. To answer this question, we newly trained and tested the network without normalizing the maps using the author-recommended contour level. Instead of using the author-recommended level, we normalized the density values of a map by using the minimum density value from the whole map as the minimum value, 0. The new results are provided in the fourth and fifth page, "phase1_nocontour" and "phase2_nocontour" in the

Supplementary Table S4 (a separate Excel file). As shown, in phase 2, the average value of all the metrics were better when the original setting with the author-recommended contour level was used. When individual maps were compared, among the 19 test maps, there are up to a few maps where the new no-contour setting performed slightly better for some metrics.

In general, I agree that some useful signal would be lost when the author-recommended contour level was used to truncate an EM map. On this dataset, overall, using author-recommended level was helpful in reducing noise. But it is case-by-case, because some maps improved by not using author-recommended level for truncation.

Therefore, for users to be able to use both versions, in Github, we provided two versions of the codes, one for maps truncated with the author-recommended level, and the other for maps not truncated by the author-recommended level.

We mentioned this results in page 18.

Results:

The authors claim that, “When only the three protein structure classes are considered, the Q3 score were 0.846 and 0.780 for 6 Å and 10 Å maps, respectively (Supplementary Table S3), which are better than results shown in Table 1 of the Emap2sec paper.” However, the authors do not report the actual values from the previous Emap2sec paper. Referring to Table 1 and Supplementary Figure 3 of their original paper reveals that the overall Emap2sec Q3 scores were actually comparable to Emap2sec+ (+/- 0.02) and in fact slightly lower in Emap2sec+ for 10A (0.780 for Emap2sec+ and 0.798 for Emap2sec).

The reviewer is right that the previous Emap2sec paper had Q3 score of 0.831 for 6A maps and 0.798 for 10A maps. In page 8, we corrected the sentence as follows: “When only the three protein structure classes are considered, the Q3 score were 0.846 and 0.780 for 6 Å and 10 Å maps, respectively (Supplementary Table S3). These values are comparable with the results shown in Table 1 of the Emap2sec papers (Q3 of 0.831 and 0.798 for 6 Å and 10 Å maps, respectively).

The authors report 106 out of 108 samples improved following phase 2 training. It would be useful to elaborate which two samples did not improve, and whether there was anything particular to set them apart.

The two maps which decreased Q4 by the phase 2 network were 1CA6 and 5GZB. Looking at Supplementary Table S3 that provides all the results of phase 1 and phase 2, the drops of the Q4 were not very large: For 1CA6, Q4 dropped by 0.0309 and 0.0273 for the 6 Å map and 10 Å map, respectively. For 5GZB, Q4 decreased by 0.00877 and 0.0247 for 6 Å and 10 Å map, respectively.

In 1CA6, for the 6 Å map, this drop was due to the drop of alpha helix recall. In this case, an original assignment of alpha helix was wrongly changed to coil. In 10 Å map, this drop was due to the recall drop of coil. Opposite to the 6 Å map case, original correct assignment of coil was changed to alpha helix. So for both cases, there are assignment changes between alpha helix and coil.

In 5GZB, for both 6 Å and 10 Å maps, the drop of Q4 was caused by recall decrease for other structures (coil). Checking the voxel-based accuracy results, the recall drop of coil occurred due to wrong revision of coil to alpha helix by the phase 2 network.

Among the 4 cases (i.e. 6 Å map and 10 Å map for 1CA6 and 5GZB), actually, for 3 cases (except for the 6 Å map of 5GZB) the overall voxel level accuracy improved by the phase 2 network. In the 6 Å map of 5GZB, the decrease of overall voxel level accuracy was 0.00499.

For both cases, the wrongly changed alpha helix structure assignment occurred at the end of an alpha helix. The change wrongly made by phase 2 was 1 to 2 residues.

We added the above explanation in page 9.

The authors do not mention the distribution of nominal resolution in experimental maps. Investigation of Supplementary Table 4 and Figure 4a reveals that low resolution maps (7-10Å) were less frequent. This could perhaps be reported in the text.

In page 12, we added a sentence to report the breakdown of the maps at different resolution: “The dataset has 6 maps between 5 to 6 Å, 7 maps between 6 to 7 Å, 1 map between 7 to 8 Å, 3 maps between 8 to 9 Å, 1 map between 9 to 10 Å, and 1 map at 10 Å.”

The authors state that “It is worthwhile to note that Emap2sec+ could detect structures from the map at 10.0 Å (EMD-8131) with Q4 of 0.575”. However, consulting the Supplementary Table 4, EMD-8131 Q4 is reported as 0.516. The authors also claim that “For the rest of the 10 maps [with resolution over 7Å], Q4 was stable around 0.45 to 0.5 regardless of the map resolution”. However, it is clear from Figure 4a and Supplementary Table 4 that a single sample was lower than that (EMDB-4671 Q4 at 0.3617). This is understandable given that the map was at 9.1Å resolution, nevertheless the results should be reported correctly in the text.

Thanks for finding these mistakes. We corrected them in page 13. For the second point for the map with resolution over 7Å, we rewrote it to “For the rest of the 10 maps with a resolution over 7Å, the average Q4 was 0.464.”

Text discussion of Figure 5 frequently reports good Q4 on presented experimental structures, however consistently omits the bad predictions (e.g. EMDB-4075 0.47 Alpha recall, EMDB-3949 0.14 Beta recall, as reported by Supplementary Table 4). This is not obvious without consulting the Supplementary Table 4 and could thus bias the reader. In some cases, these lower scores are perhaps understandable given the small proportion of the particular class represented in the structure (e.g. beta-strands in EMDB-3949), nevertheless the negative results should still be reported in text while discussing the particular structure, as this largely determines how useful the method is for experimental data.

In the protein structures of EMDB-4075 (PDB ID: 5LMP), there are 793 alpha helix residues. Among them, 379 residues were correctly identified as alpha helix (47%) but 282 were incorrectly recognized as other structures (loop), 118 as beta strands, and 14 as DNA/RNA. Thus, the largest number of misassignment for alpha helix residues occurred to other structures. This is probably because other structure is the largest class (1290 residues) among the three protein classes in this map and also a helix is next to loop regions. Since EMDB-4075 was shown in Figure 5b, we added this explanation in the section.

EMDB-3949 has only 34 beta strand residues and all of them are isolated as 2 residue-long beta strands. These beta strands do not form beta sheets with another beta strand, and actually, these residues are not even recognized as beta strand if a different secondary structure definition program was used. Thus, it was difficult to recognize these residues as beta strand by the neural network, and maybe not unreasonable not to recognize them as beta strand. We mentioned this at page 15.

Comparison with related works:

Stating that Haruspex did not perform well on simulated maps is inadequate and such comparison should not be made. The fact that it did not perform well for simulated data is obvious by design, as Haruspex was trained for experimental maps. The Supplementary Figure 3 shows the comparison between Haruspex on both 6 and 10A simulated maps against Emap2sec+, which, as earlier stated, was trained separately for simulated and experimental data (and even made the distinction between 6 and 10A maps in simulated data). These comparisons should be excluded from the paper.

We have removed the Haruspex results for the simulated data from the Supplementary Figure 5.

The comparison made between Haruspex and Emap2sec+ on experimental data is far more sensible. However, given the fact that Haruspex was trained for high (sub 4A) resolution data, it is interesting to see that many Haruspex predictions were only slightly improved on by Emap2sec+, and in some cases Haruspex was actually better. While the large majority of predictions by Emap2sec+ were indeed better than Haruspex, some of these exceptions could be discussed in text. Also, it would be interesting to read a more comprehensive comparison between the two methods, e.g. average difference in accuracy or average per-class difference in accuracy. For example, from the figure it appears that loops ('others' category) were on average classified much better by Emap2sec+, but for DRNA only 10 out of 19 structures (about half) were better with Emap2sec+ (although I can only count 18 DNA/RNA data points in that figure rather than 19). The text only states "(...) Haruspex did not work well on our low resolution datasets of simulated maps and experimental maps (...)", which is overly dismissive and performs an inadequate comparison.

I agree that we should have done more careful analysis. We have totally rewritten the section. You are right that there are cases that Haruspex did better than Emap2sec+. We compared the average values of the classes and also discussed individual structure class separately. The revised section is in page 16.

REVIEWERS' COMMENTS

Reviewer #2 (Remarks to the Author):

The authors have now revised their original manuscript, providing considerably more details on the architecture, data sampling approach, inference times, class imbalance, and comparison with related work. The authors have also performed additional analysis to address some of the comments, including comparison without author recommended threshold normalisation of maps, binary Q2 analysis (protein vs. D/RNA), and analysis with a single multi-class classifier (rather than 4 binary plus 1 multi-class). Overall, the paper clarity has much improved following this revision. It is interesting to note that rephrasing the problem as a binary classification (Q2) between protein and nucleic acid has considerably improved the performance, which perhaps suggests that the method is much more useful in that context from the user's perspective. The authors also elaborated more on some of the negative results and illustrated the reasons behind them, and corrected some inconsistencies between the manuscript text and Supplementary Tables in this work as well as the previous iteration of this work. The author's comparison with related work now contains much more detail, including the isolated cases where related work performed better. As a very minor correction, it would be useful to see memory consumption details in addition to the time complexity. Also, very minor grammar errors still occur in the manuscript text.

LANGUAGE AND GRAMMAR:

The quality of the language has much improved, but some grammar issues and typos still exist, mainly in the newly added text sections. Some examples below.

- 1) P8, last sentence of paragraph 2: "The change made incorrectly by the phase 2 network fors a map (...)." instead of "for a map".
- 2) Figure 3, last sentence: "Accuracies for RNA was: (...)." instead of "were".
- 3) P11: "The large volume RNA structures in this map was well identified at Q4 (...)" instead of "were".
- 4) P18, first sentence of paragraph 4: "The input density data for Emap2sec+ was voxels of size of 11 3 Å 3" instead of "were".

DETAILED COMMENTS WITH REGARDS TO EACH REVISION:

1) The authors have now addressed in the paper their rationale behind phrasing the problem as a classification and mentioned that it could also be phrased as a segmentation and addressed with alternative architectures. However, as a minor correction, in their response letter (but not in the manuscript) the authors mention that the main focus of segmentation is detecting foreground vs. background, therefore rendering it less useful for application in their low resolution data. This is not factually correct, as multi-class segmentation applications exists in many areas of computer vision and outside of it (including life sciences and Haruspex is one example of such segmentation, where the distinction is made between alpha-helices, beta-sheets and other structures rather than just background/foreground or protein/not-protein classes). For reference, see examples of SegNet which is a multi-class segmentation network applied to the task of self-driving cars as a prime example of such approaches. However, there is no need to address this comment further in the paper revisions as this is only remarking on a comment from the author's response letter. The author's manuscript revision in this regard outlining their reasons and listing alternate approaches is a satisfactory response.

2) The authors have now remarked on the computational (time) performance of inference with respect to map size and number of voxels. In addition to computational costs outlined by the authors in Supplementary Figure S2, it would be interesting to see the memory consumption, either as a figure, or at the least as a short text remark. This is important from the user perspective - can anyone run this on their laptop or do they need a cluster? How much RAM/GPU memory do they need to make an inference on a single map?

3) Following recommendation, the authors have now added a set of new results following analysis

of the network performance for a binary (Q2) segmentation of protein vs. D/RNA (i.e. without the secondary structure prediction Q4). The results show much higher accuracy for this type of analysis, suggesting that the network is much more useful in this context rather than 4-class classification. It is also interesting that in some cases D/RNA accuracy has actually dropped a little in this setting (e.g. Fig. 4c, third sample from the right at 8.6-8.8) which could be addressed in text, but it's optional.

4) The authors have addressed the issue of class imbalance, which was handled by over-sampling, and regions containing background (as a central voxel) were excluded. No further need to address this.

5) The author's provided an improved explanation of the network architecture, data sampling and network input sizes. Their new explanation clarifies why the network inputs were described in Angstroms and how they relate to grid sizes. The vocabulary is still confusing and contrary to the standards in computer vision / machine learning, where voxel is typically reserved to describe a discrete grid point in a 3D image (analogous to pixels), but authors remain consistent in their use of the two terms ("grid point" and "voxel") which makes it much more clear. There is no further need to address this.

6) The authors provide more explanation regarding their choice of data split, including description of how over-sampling was used. In their response letter they agree that such an ensemble approach is useful to prevent overfitting. Optionally, it would be useful to mention this in text.

7) In the original manuscript the authors used recommended threshold level to normalise the maps, which has the potential to truncate some side-chains or mobile areas of the protein. To address this, the authors performed a new analysis without the use of the author recommended threshold value. The results improved for some maps, but not for others and the authors note this should be considered on a case-by-case basis. The authors enabled this option in the code for the user's individual consideration, which is very useful. There is no further need to address this.

8) The authors added more information on the breakdown of nominal resolution in experimental maps. There is no further need to address this.

9) The authors corrected numerical inconsistencies between the manuscript text and both, Supplementary Material as well as previous iteration of this work. There is no further need to address this.

10) The authors have revised the comparison with related work (Haruspex). The comparison with synthetic data was removed, as Haruspex was trained specifically for experimental structures. The comparison with experimental structures now contains more information describing the differences in detail, as well as some of the isolated exceptions where Haruspex performed better. Overall average per-voxel and per-residue comparison was also included. There is no further need to address this.

Responses to comments by Reviewer #2:

The authors have now revised their original manuscript, providing considerably more details on the architecture, data sampling approach, inference times, class imbalance, and comparison with related work. The authors have also performed additional analysis to address some of the comments, including comparison without author recommended threshold normalization of maps, binary Q2 analysis (protein vs. D/RNA), and analysis with a single multi-class classifier (rather than 4 binary plus 1 multi-class). Overall, the paper clarity has much improved following this revision. It is interesting to note that rephrasing the problem as a binary classification (Q2) between protein and nucleic acid has considerably improved the performance, which perhaps suggests that the method is much more useful in that context from the user's perspective. The authors also elaborated more on some of the negative results and illustrated the reasons behind them, and corrected some inconsistencies between the manuscript text and Supplementary Tables in this work as well as the previous iteration of this work. The author's comparison with related work now contains much more detail, including the isolated cases where related work performed better. As a very minor correction, it would be useful to see memory consumption details in addition to the time complexity. Also, very minor grammar errors still occur in the manuscript text.

We really appreciate the reviewer's constructive comments and guidance.

LANGUAGE AND GRAMMAR:

The quality of the language has much improved, but some grammar issues and typos still exist, mainly in the newly added text sections. Some examples below.

- 1) P8, last sentence of paragraph 2: "The change made incorrectly by the phase 2 network for a map (...)." instead of "for a map".
- 2) Figure 3, last sentence: "Accuracies for RNA was: (...)." instead of "were".
- 3) P11: "The large volume RNA structures in this map was well identified at Q4 (...)" instead of "were".
- 4) P18, first sentence of paragraph 4: "The input density data for Emap2sec+ was voxels of size of 11 3 Å 3" instead of "were".

We have fixed those typos in the updated version.

DETAILED COMMENTS WITH REGARDS TO EACH REVISION:

1) The authors have now addressed in the paper their rationale behind phrasing the problem as a classification and mentioned that it could also be phrased as a segmentation and addressed with alternative architectures. However, as a minor correction, in their response letter (but not in the manuscript) the authors mention that the main focus of segmentation is detecting foreground vs. background, therefore rendering it less useful for application in their low resolution data. This is not factually correct, as multi-class segmentation applications exists in many areas of computer vision and outside of it (including life sciences and Haruspex is one example of such segmentation, where the distinction is made between alpha-helices, beta-sheets and other structures rather than just background/foreground or protein/not-protein classes). For reference, see examples of SegNet which is a multi-class segmentation network applied to the task of self-driving cars as a prime example of such approaches. However, there is no need to address this comment further in the paper revisions as this is only remarking on a comment from the author's response letter. The author's manuscript revision in this regard outlining their reasons and listing alternate approaches is a satisfactory response.

Thank you very much. We will check the SegNet paper.

2) The authors have now remarked on the computational (time) performance of inference with respect to map size and number of voxels. In addition to computational costs outlined by the authors in Supplementary Figure S2, it would be interesting to see the memory consumption, either as a figure, or at the least as a short text remark. This is important from the user perspective - can anyone run this on their laptop or do they need a cluster? How much RAM/GPU memory do they need to make an inference on a single map?

We added the GPU memory usage in the new Supplementary Table 2. Emap2sec+ can run on a laptop computer with GPU with a small memory size of even less than 10GB. Currently CPU is not supported but we planned to support it in very near future. We mentioned this table in the main text (page 5).

3) Following recommendation, the authors have now added a set of new results following analysis of the network performance for a binary (Q2) segmentation of protein vs. D/RNA (i.e. without the secondary structure prediction Q4). The results show much higher accuracy for this type of analysis, suggesting that the network is much more useful in this context rather than 4-class classification. It is also interesting that in some cases D/RNA accuracy has actually dropped a little in this setting (e.g. Fig. 4c, third sample from the right at 8.6-8.8) which could be addressed in text, but it's optional.

We added the following sentence in the Figure 4 caption:

“Note that values for DNA/RNA can be different from panel a, which reports the results of four-class classification. Since the probability of the protein class was computed as the sum of probabilities of three secondary structure classes, a DNA/RNA assignment in the four-class classification can be changed to protein in the binary classification.”

If an original assignment is DNA/RNA with a small margin of the probability, it can be altered to protein in the binary classification evaluation we did here if the sum of probabilities of alpha helix, beta strand, coil exceed the DNA/RNA probability.

4) The authors have addressed the issue of class imbalance, which was handled by over-sampling, and regions containing background (as a central voxel) were excluded. No further need to address this.

Thank you.

5) The author's provided an improved explanation of the network architecture, data sampling and network input sizes. Their new explanation clarifies why the network inputs were described in Angstroms and how they relate to grid sizes. The vocabulary is still confusing and contrary to the standards in computer vision / machine learning, where voxel is typically reserved to describe a discrete grid point in a 3D image (analogous to pixels), but authors remain consistent in their use of the two terms (“grid point” and “voxel”) which makes it much more clear. There is no further need to address this.

Thank you.

6) The authors provide more explanation regarding their choice of data split, including description of how over-sampling was used. In their response letter they agree that such an ensemble approach is useful to prevent overfitting. Optionally, it would be useful to mention this in text.

We added mentioned the ensemble approach in page 4, mentioning the ensemble approach of the binary and multi-class classifier performed better than simply using the multi-class classifier.

7) In the original manuscript the authors used recommended threshold level to normalize the maps, which has the potential to truncate some side-chains or mobile areas of the protein. To address this, the authors performed a new analysis without the use of the author recommended threshold value. The results improved for some maps, but not for others and the authors note this should be considered on a case-by-case basis. The authors enabled this option in the code for the user's individual consideration, which is very useful. There is no further need to address this.

Thank you.

8) The authors added more information on the breakdown of nominal resolution in experimental maps. There is no further need to address this.

Thank you.

9) The authors corrected numerical inconsistencies between the manuscript text and both, Supplementary Material as well as previous iteration of this work. There is no further need to address this.

Thank you.

10) The authors have revised the comparison with related work (Haruspex). The comparison with synthetic data was removed, as Haruspex was trained specifically for experimental structures. The comparison with experimental structures now contains more information describing the differences in detail, as well as some of the isolated exceptions where Haruspex performed better. Overall average per-voxel and per-residue comparison was also included. There is no further need to address this.

Thank you.